

# Identifying accurate metagenome and amplicon software via a meta-analysis of sequence to taxonomy benchmarking studies

Paul P. Gardner[1,2], Renee J. Watson[1], Xochitl C. Morgan[3], Jenny L. Draper[4], Robert D. Finn[5], Sergio E. Morales[3] and Matthew B. Stott[1]

[1] Biomolecular Interactions Centre, School of Biological Sciences, University of Canterbury, Christchurch, New Zealand
[2] Department of Biochemistry, University of Otago, Dunedin, New Zealand
[3] Department of Microbiology and Immunology, University of Otago, Dunedin, New Zealand
[4] Institute of Environmental Science and Research, Porirua, New Zealand
[5] European Molecular Biology Laboratory, European Bioinformatics Institute (EMBL-EBI), Cambridge, UK

## ABSTRACT

Metagenomic and meta-barcode DNA sequencing has rapidly become a widely-used technique for investigating a range of questions, particularly related to health and environmental monitoring. There has also been a proliferation of bioinformatic tools for analysing metagenomic and amplicon datasets, which makes selecting adequate tools a significant challenge. A number of benchmark studies have been undertaken; however, these can present conflicting results. In order to address this issue we have applied a robust $Z$-score ranking procedure and a network meta-analysis method to identify software tools that are consistently accurate for mapping DNA sequences to taxonomic hierarchies. Based upon these results we have identified some tools and computational strategies that produce robust predictions.

## INTRODUCTION

Metagenomics, meta-barcoding and related high-throughput environmental DNA (eDNA) or microbiome sequencing approaches have accelerated the discovery of small and large scale interactions between ecosystems and their biota. Metagenomics is frequently used as a catch-all term to cover metagenomic, meta-barcoding and other environmental DNA sequencing methods, which we will also apply apply. The application of these methods has advanced our understanding of microbiomes, disease, ecosystem function, security and food safety (*Cho & Blaser, 2012*; *Baird & Hajibabaei, 2012*; *Bohan et al., 2017*). The classification of DNA sequences can be broadly divided into amplicon (barcoding) and genome-wide (metagenome) approaches. The amplicon, or barcoding, -based approaches target genomic marker sequences such as ribosomal RNA genes (*Woese, 1987*; *Woese, Kandler & Wheelis, 1990*; *Hugenholtz & Pace, 1996*; *Tringe & Hugenholtz, 2008*) (16S, 18S, mitochondrial 12S), RNase P RNA (*Brown et al., 1996*), or internal transcribed spacers

Corresponding author
Paul P. Gardner,
paul.gardner@otago.ac.nz

(ITS) between ribosomal RNA genes (*Schoch et al., 2012*). These regions are amplified from extracted DNA by PCR, and the resulting DNA libraries are sequenced. In contrast, genome-wide, or metagenome, -based approaches sequence the entire pool of DNA extracted from a sample with no preferential targeting for particular markers or taxonomic clades. Both approaches have limitations that influence downstream analyses. For example, amplicon target regions may have unusual DNA features (e.g., large insertions or diverged primer annealing sites), and consequently these DNA markers may fail to be amplified by PCR (*Brown et al., 2015*). While the metagenome-based methods are not vulnerable to primer bias, they may fail to detect genetic signal from low- abundance taxa if the sequencing does not have sufficient depth, or may under-detect sequences with a high G+C bias (*Tringe et al., 2005*; *Ross et al., 2013*)

High-throughput sequencing (HTS) results can be analysed using a number of different strategies (Fig. S1) (*Thomas, Gilbert & Meyer, 2012*; *Sharpton, 2014*; *Oulas et al., 2015*; *Quince et al., 2017*). The fundamental goal of many of these studies is to assign taxonomy to sequences as specifically as possible, and in some cases to cluster highly-similar sequences into ''operational taxonomic units'' (OTUs) (*Sneath & Sokal, 1963*). For greater accuracy in taxonomic assignment, metagenome and amplicon sequences may be assembled into longer ''contigs'' using any of the available sequence assembly tools (*Olson et al., 2017*; *Breitwieser, Lu & Salzberg, 2017*). The reference-based methods (also called ''targeted gene assembly'') make use of conserved sequences to constrain sequence assemblies. These have a number of reported advantages including reducing chimeric sequences, and improving the speed and accuracy of assembly relative to *de novo* methods (*Zhang, Sun & Cole, 2014*; *Wang et al., 2015*; *Huson et al., 2017*; *Nurk et al., 2017*).

Metagenomic sequences are generally mapped to a reference database of sequences labelled with a hierarchical taxonomic classification. The level of divergence, distribution and coverage of mapped taxonomic assignments allows an estimate to be made of where the sequence belongs in the established taxonomy . This is commonly performed using the lowest common ancestor approach (LCA) (*Huson et al., 2007*). Some tools, however, avoid this computationally-intensive sequence similarity estimation, and instead use alignment-free approaches based upon sequence composition statistics (e.g., nucleotide or k-mer frequencies) to estimate taxonomic relationships (*Gregor et al., 2016*).

In this study we identified seven published evaluations, of tools that estimate taxonomic origin from DNA sequences (*Bazinet & Cummings, 2012*; *Peabody et al., 2015*; *Lindgreen, Adair & Gardner, 2016*; *Siegwald et al., 2017*; *McIntyre et al., 2017*; *Sczyrba et al., 2017*; *Almeida et al., 2018*). Of these, four evaluations met our criteria for a neutral comparison study (*Boulesteix, Lauer & Eugster, 2013*) (see Table S1). These are summarised in Table 1 (*Bazinet & Cummings, 2012*; *Lindgreen, Adair & Gardner, 2016*; *Siegwald et al., 2017*; *Almeida et al., 2018*) and include accuracy estimates for 25 eDNA classification tools. We have used network meta-analysis techniques and non-parametric tests to reconcile variable and sometimes conflicting reports from the different evaluation studies. Our result is a short list of methods that have been consistently reported to produce accurate interpretations of metagenomics results. This study reports one of the first meta-analyses

**Table 1  A summary of the main features of the four software evaluations used for this study, including the positive controls employed (the sources of sequences from organisms with known taxonomic placements, whether negative control sequences were used, the approaches for excluding reference sequences from the positive control sequences, and the metrics that were collected for tool evaluation.** The accuracy measures are defined in Table 2, the abbreviations used above are Matthews Correlation Coefficient (MCC), Negative Predictive Value (NPV), Positive Predictive Value (PPV), Sensitivity (Sen), Specificity (Spec).

| Paper | Positive controls | Negative controls | Reference exclusion method | Metrics |
|---|---|---|---|---|
| *Almeida et al. (2018)* | 12 *in silico* mock communities from 208 different genera. | – | 2% of positions "randomly mutated" | Sequence Level (Sen., *F*-measure) |
| *Bazinet & Cummings (2012)* | Four published *in silico* mock communities from 742 taxa (*Stranneheim et al., 2010*; *Liu et al., 2010*; *Patil et al., 2011*; *Gerlach & Stoye, 2011*) | – | – | Sequence Level (Sen., PPV) |
| *Lindgreen, Adair & Gardner (2016)* | Six *in silico* mock communities from 417 different genera. | Shuffled sequences | Simulated evolution | Sequence Level (Sen., Spec., PPV, NPV, MCC) |
| *Siegwald et al. (2017)* | 36 *in silico* mock communities from 125 bacterial genomes. | – | – | Sequence Level (Sen, PPV, *F*-measure) |

of neutral comparison studies, fulfilling the requirement for an apex study in the evidence pyramid for benchmarking (*Boulesteix, Wilson & Hapfelmeier, 2017*).

## Overview of environmental DNA classification evaluations

Independent **benchmarking of bioinformatic software** provides a valuable resource for determining the relative performance of software tools, particularly for problems with an overabundance of tools. Some established criteria for reliable benchmarks are: **1**. The main focus of the study should be the evaluation and not the introduction of a new method; **2**. the authors should be reasonably neutral (i.e., not involved in the development of methods included in an evaluation); and **3**. the test data, evaluation and methods should be selected in a rational way (*Boulesteix, Lauer & Eugster, 2013*). Criteria 1 and 2 are straightforward to determine, but criterion 3 is more difficult to evaluate as it includes identifying challenging datasets and appropriate metrics for accurate accuracy reporting (*Boulesteix, 2010*; *Jelizarow et al., 2010*; *Norel, Rice & Stolovitzky, 2011*). Based upon our literature reviews and citation analyses, we have identified seven published evaluations of eDNA analysis, we have assessed these against the above three principles and four of these studies meet the inclusion criteria (assessed in Table S1) (*Bazinet & Cummings, 2012*; *Lindgreen, Adair & Gardner, 2016*; *Siegwald et al., 2017*; *Almeida et al., 2018*). These studies are summarised in Table 1.

In the following sections we discuss issues with collecting trusted datasets, including the selection of positive and negative control data that avoid datasets upon which methods may have been over-trained. We describe measures of accuracy for predictions and describe the characteristics of ideal benchmarks, with examples of published benchmarks that meet these criteria.

## Positive and negative control dataset selection

The selection of datasets for evaluating software can be a significant challenge due to the need for these to be independent of past training datasets, reliable, well-curated, robust and representative of the large population of all possible datasets (*Boulesteix, Wilson & Hapfelmeier, 2017*). **Positive control** datasets can be divided into two different strategies, namely the *in vitro* and *in silico* approaches for generating mock communities.

*In vitro* methods involve generating microbial consortia in predetermined ratios of microbial strains, extracting the consortium DNA, sequencing and analysing these using standard eDNA pipelines (*Jumpstart Consortium Human Microbiome Project Data Generation Working Group, 2012*; *Singer et al., 2016b*). Non-reference sequences can also be included to this mix as a form of negative control. The accuracy of the genome assembly, genome partitioning (binning) and read depth proportional to consortium makeup can then be used to confirm software accuracy. In principle, every eDNA experiment could employ *in vitro* positive and negative controls by "spiking" known amounts of DNA from known sources, as has been widely used for gene expression analysis (*Yang, 2006*) and increasingly for eDNA experiments (*Bowers et al., 2015*; *Singer et al., 2016a*; *Hardwick et al., 2018*).

*In silico* methods use selected publicly-available genome sequences. Simulated metagenome sequences can be derived from these (*Richter et al., 2008*; *Huang et al., 2012*; *Angly et al., 2012*; *Caboche et al., 2014*). It is important to note that ideally-simulated sequences are derived from species that are **not present** in established reference databases, as this is a more realistic simulation of most eDNA surveys. A number of different strategies have been used to control for this (*Peabody et al., 2015*; *Lindgreen, Adair & Gardner, 2016*; *Sczyrba et al., 2017*). *Peabody et al. (2015)* used "clade exclusion", in which sequences used for an evaluation are removed from reference databases for each software tool. Lindgreen et al. used "simulated evolution" to generate simulated sequences of varying evolutionary distances from reference sequences; similarly, *Almeida et al. (2018)* simulated random mutations for 2% of nucleotides in each sequence. *Sczyrba et al. (2017)* restricted their analysis to sequences sampled from recently-deposited genomes, increasing the chance that these are not included in any reference databases. These strategies are illustrated in Fig. 1.

Another important consideration is the use of **negative controls**. These can be randomised sequences (*Lindgreen, Adair & Gardner, 2016*), or from sequence not expected to be found in reference databases (*McIntyre et al., 2017*). The resulting negative-control sequences can be used to determine false-positive rates for different tools. We have summarised the positive and negative control datasets from various published software evaluations in Table 1, along with other features of different evaluations of DNA classification software.

## Metrics used for software benchmarking

The metrics used to evaluate software play an important role in determining the fit for different tasks. For example, if a study is particularly interested in identifying rare species in samples, then a method with a high true-positive rate (also called **sensitivity** or **recall**)
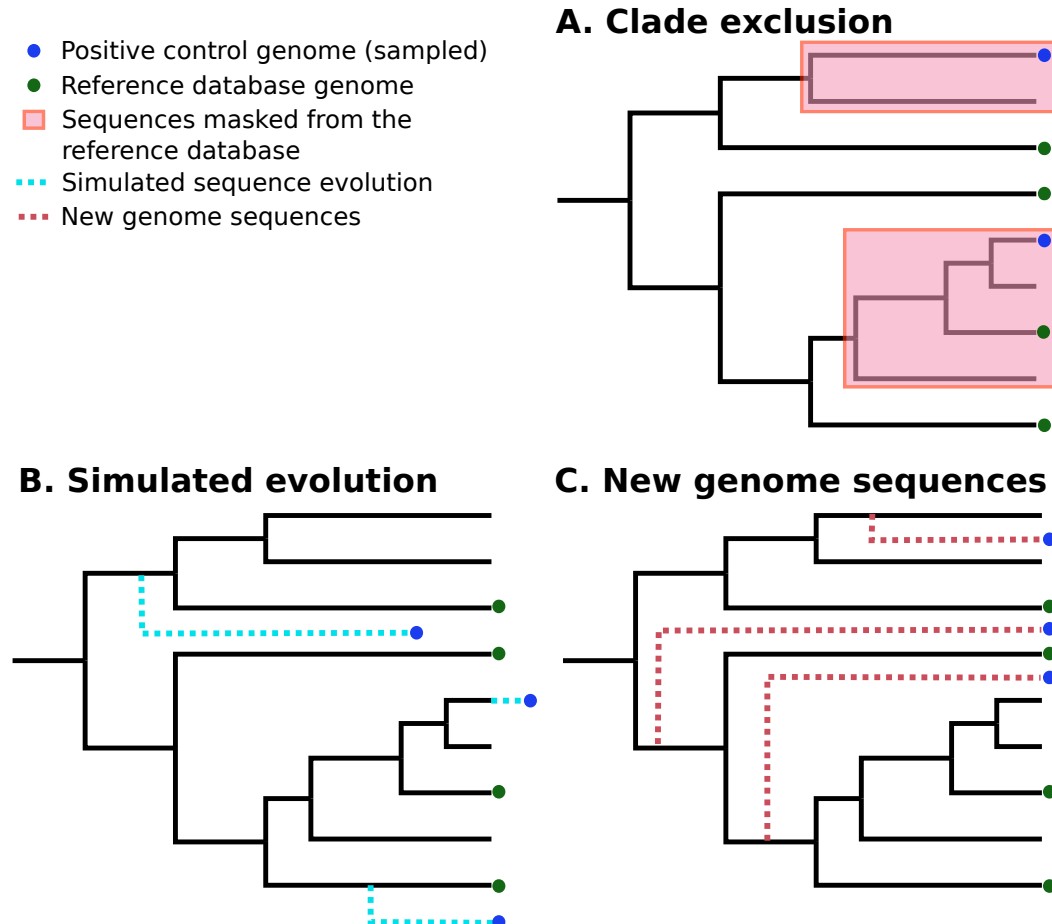

**Figure 1 Three different strategies for generating positive control sequencing datasets.** Three different strategies for generating positive control sequencing datasets, i.e., genome/barcoding datasets of known taxonomic placement that are absent from existing reference databases. These are: (A) "clade exclusion", where positive control sequences are selectively removed from reference databases (*Peabody et al., 2015*); (B) "simulated evolution", where models of sequence evolution are used to generate sequences of defined divergence times from any ancestral sequence or location on a phylogenetic tree e.g., (*Stoye, Evers & Meyer, 1998*; *Dalquen et al., 2012*); and (C) "new genome sequences" are genome sequences that have been deposited in sequence archives prior to the generation of any reference sequence database used by analysis tools (*Sczyrba et al., 2017*) .

may be preferable. Conversely, for some studies, false positive findings may be particularly detrimental, in which case a good true positive rate may be sacrificed in exchange for a lowering the false positive rate. Some commonly used measures of accuracy, including *sensitivity* (recall/true positive accuracy), *specificity* (true negative accuracy) and *F-measure* (the trade-off between recall and precision) are summarised in Table 2.

The definitions of "true positive", "false positive", "true negative" and "false negative" (TP, FP, TN and FN respectively) are also an important consideration. There are two main ways this has been approached, namely per-sequence assignment and per-taxon assignment. Estimates of per-sequence accuracy values can be made by determining whether

**Table 2  Some commonly used measures of "accuracy" for software predictions.** These are dependent upon counts of true positives (TP), false positives (FP), true negatives (TN) and false negatives (FN) which can be computed from comparisons between predictions and ground-truths (*Lever, Krzywinski & Altman, 2016*).

$Sensitivity = \frac{TP}{TP+FN}$
(a.k.a. recall, true positive rate)

$Specificity = \frac{TN}{TN+FP}$
(a.k.a. true negative rate)

$PPV = \frac{TP}{TP+FP}$
(a.k.a. positive predictive value, precision, sometimes mis-labelled "specificity")

$F\text{-}measure = \frac{2*Sensitivity*PPV}{Sensitivity+PPV}$
$F\text{-}measure = \frac{2TP}{2TP+FP+FN}$
(a.k.a. F1 score)

$Accuracy = \frac{TP+TN}{TP+TN+FP+FN}$

$FPR = \frac{FP}{FP+TN}$
(a.k.a false positive rate)

individual sequences were correctly assigned to a particular taxonomic rank (*Peabody et al., 2015*; *Lindgreen, Adair & Gardner, 2016*; *Siegwald et al., 2017*). Alternatively, per-taxon accuracies can be determined by comparing reference and predicted taxonomic distributions (*Sczyrba et al., 2017*). The per-taxon approach may lead to erroneous accuracy estimates as sequences may be incorrectly assigned to included taxa. Cyclic-errors can then cancel, leading to inflated accuracy estimates. However, per-sequence information can be problematic to extract from tools that only report profiles.

**Successfully** recapturing the frequencies of different taxonomic groups as a **measure of community diversity** is a major aim for eDNA analysis projects. There have been a variety of approaches for quantifying the accuracy of this information. Pearson's correlation coefficient (*Bazinet & Cummings, 2012*), L1-norm (*Sczyrba et al., 2017*), the sum of absolute log-ratios (*Lindgreen, Adair & Gardner, 2016*), the log-modulus (*McIntyre et al., 2017*) and the Chao 1 error (*Siegwald et al., 2017*) have each been used. This lack of consensus has made comparing these results a challenge.

The amount of variation between the published benchmarks, including varying taxonomies, taxonomic levels and whether sequences or taxa were used for evaluations can also impede comparisons between methods and the computation of accuracy metrics. To illustrate this we have summarised the variation of *F*-measures (a measure of accuracy) between the four benchmarks we are considering in this work (Fig. 2).

## METHODS

**Literature search:** In order to identify benchmarks of metagenomic and amplicon software methods, an initial list of publications was curated. Further literature searches and trawling of citation databases (chiefly Google Scholar) identified a comprehensive list of seven evaluations (Table 1), in which "*F*-measures" were either directly reported, or could be computed from Supplemental Information 1. These seven studies were then evaluated against the three principles of benchmarking (*Boulesteix, Lauer & Eugster, 2013*), four studies meeting all three principles and were included in the subsequent analyses (see Table S1 for details).

A list of published eDNA classification software was curated manually. This made use of a community-driven project led by (*Jacobs, 2017*). The citation statistics for each software

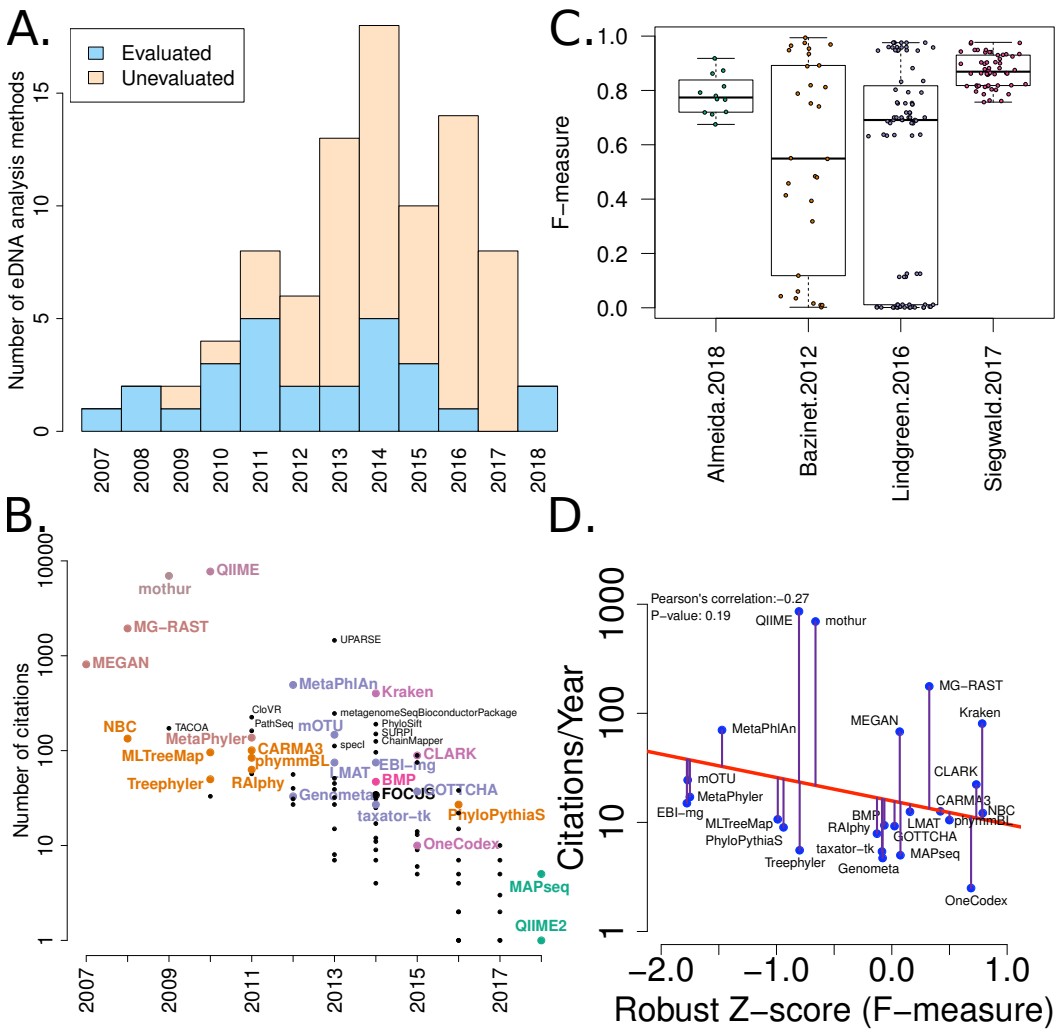

**Figure 2** **Summary of software tools, publication dates, citations and benchmarks.** More than 80 metagenome classification tools have been published in the last 10 years (Jacobs). A fraction of these (29%) have been independently evaluated. (B) The number of citations for each software tool versus the year it was published. Software tools that have been evaluated are coloured and labelled (using colour combinations consistent with evaluation paper(s), see right). Those that have not been evaluated, yet have been cited >100 times are labelled in black. (C) Box-whisker plots illustrating the distributions of accuracy estimates based upon reported *F*-measures using values from 4 different evaluation manuscripts (*Bazinet & Cummings, 2012*; *Lindgreen, Adair & Gardner, 2016*; *Siegwald et al., 2017*; *Almeida et al., 2018*). (D) The relationship between publication citation counts and the corresponding tool accuracy estimate, as measured by a normalised *F*-measure (see 'Methods' for details).

publication were manually collected from Google Scholar (in July 2017). These values were used to generate Fig. 2.

**Data extraction:** Accuracy metrics were collected from published datasets using a mixture of manual collection from Supplemental Information 1 and automated harvesting of data from online repositories. For a number of the benchmarks, a number of non-independent accuracy estimates were taken, for example different parameters, reference

databases or taxonomic levels were used for the evaluations. We have combined all non-independent accuracy measurements using a median value, leaving a single accuracy measures for each tool and benchmark dataset combination. The data, scripts and results are available from: https://github.com/Gardner-BinfLab/meta-analysis-eDNA-software.

**Data analysis:** Each benchmark manuscript reports one or more $F$-measures for each software method. Due to the high variance of $F$-measures between studies (see Fig. 2C and Fig. S3 for a comparison), we renormalised the $F$-measures using the following formula:

$$Robust\ Z\ score = \frac{x_i - median(X)}{mad(X)}$$

Where the "$mad$" function is the median absolute deviation, "$X$" is a vector containing all the $F$-measures for a publication and "$x_i$" is each $F$-measure for a particular software tool. Robust $Z$-scores can then be combined to provide an overall ranking of methods that is independent of the methodological and data differences between studies (Fig. 3). The 95% confidence intervals for median robust $Z$-scores shown in Fig. 3 were generated using 1,000 bootstrap resamplings from the distribution of values for each method, extreme ($F = \{0, 1\}$) values seeded into each $X$ in order to capture the full range of potential $F$-measures.

Network meta-analysis was used to provide a second method that accounts for differences between studies. We used the "netmeta" and "meta" software packages to perform the analysis. As outlined in Chapter 8 of the textbook "Meta-Analysis with R", (*Schwarzer, Carpenter & Rücker, 2015*), the metacont function with Hedges' G was used to standardise mean differences and estimate fixed and random effects for each method within each benchmark. The 'netmeta' function was then used to conduct a pairwise meta-analysis of treatments (tools) across studies. This is based on a graph-theoretical analysis that has been shown to be equivalent to a frequentists network meta-analysis (*Rücker, 2012*). The 'forest' function was used on the resulting values to generate Fig. 4A.

## Review of results

We have mined independent estimates of sensitivity, positive predictive values (PPV) and $F$-measures for 25 eDNA classification tools, from three published software evaluations. A matrix showing presence-or-absence of software tools in each publication is illustrated in Fig. 3A. Comparing the list of 25 eDNA classification tools to a publicly available list of eDNA classification tools based upon literature mining and crowd-sourcing, we found that 29% (25/88) of all published tools have been evaluated in the four of seven studies we have identified as neutral comparison studies (details in Table S1) (*Boulesteix, Lauer & Eugster, 2013*). The unevaluated methods are generally recently published (and therefore have not been evaluated yet) or may no longer be available, functional, or provide results in a suitable format for evaluation (see Fig. 2A). Several software tools have been very widely cited (Fig. 2B), yet caution must be used when considering citation statistics, as the number of citations is not correlated with accuracy (Fig. 2D) (*Lindgreen, Adair & Gardner, 2016; Gardner et al., 2017*). For example, the tools that are published early are more likely to be widely cited, or it may be that some articles are not necessarily cited for the software tool. For example, the MEGAN1 manuscript is often cited for one of the first implementations of

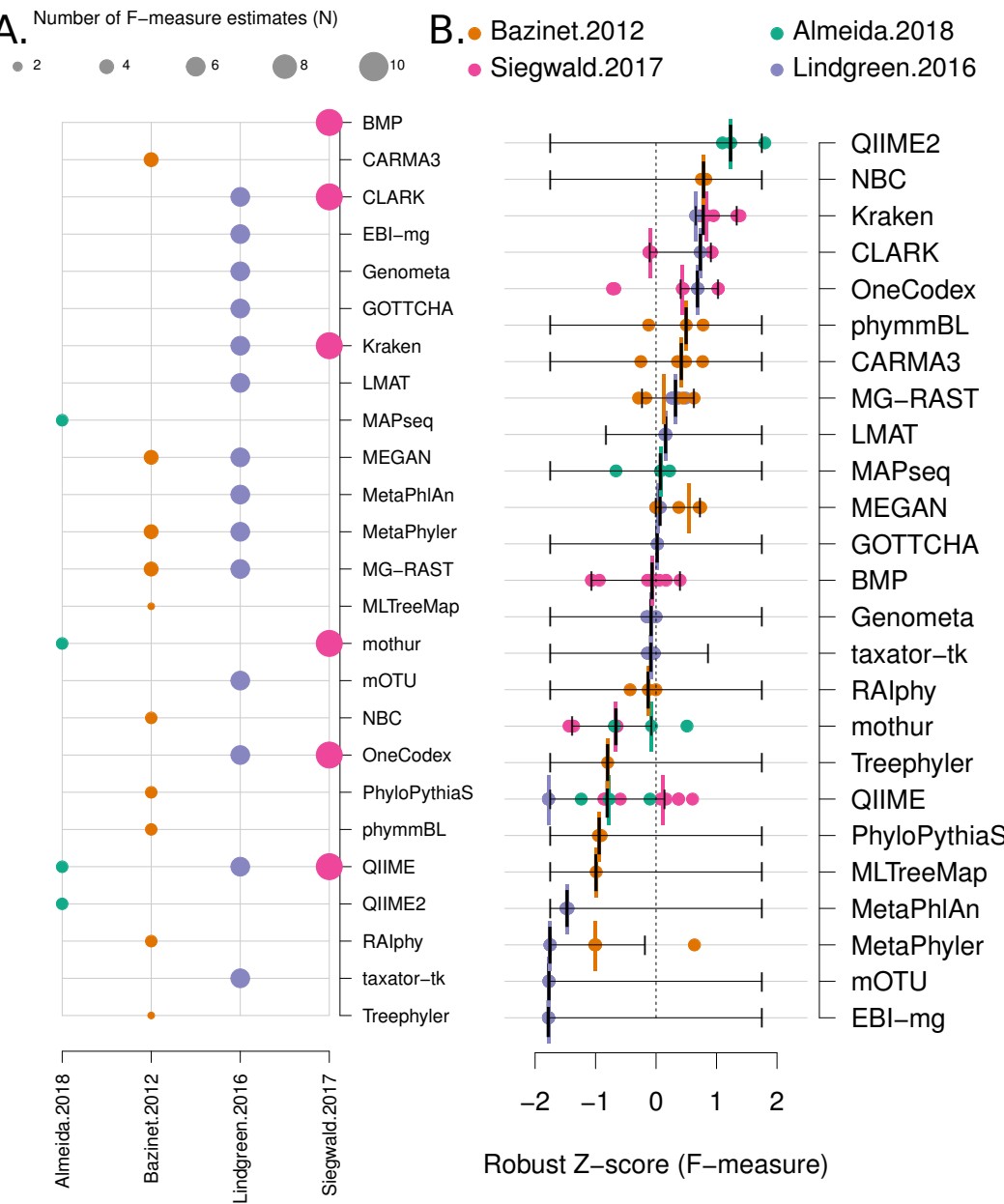

**Figure 3  Robust *Z* score based comparison of software tools.** (A) A matrix indicating metagenome analysis tools in alphabetical order (named on the right axis) versus a published benchmark on the bottom axis. The circle size is proportional to the number of *F*-measure estimates from each benchmark. (B) a ranked list of metagenome classification tools. The median *F*-measure for each tool is indicated with a thick black vertical line. Bootstrapping each distribution (seeded with the extremes from the interval) 1,000 times, was used to determine a 95% confidence interval for each median. These are indicated with thin vertical black lines. Each *F*-measure for each tool is indicated with a coloured point, colour indicates the manuscript where the value was sourced. Coloured vertical lines indicate the median *F*-measure for each benchmark for each tool.

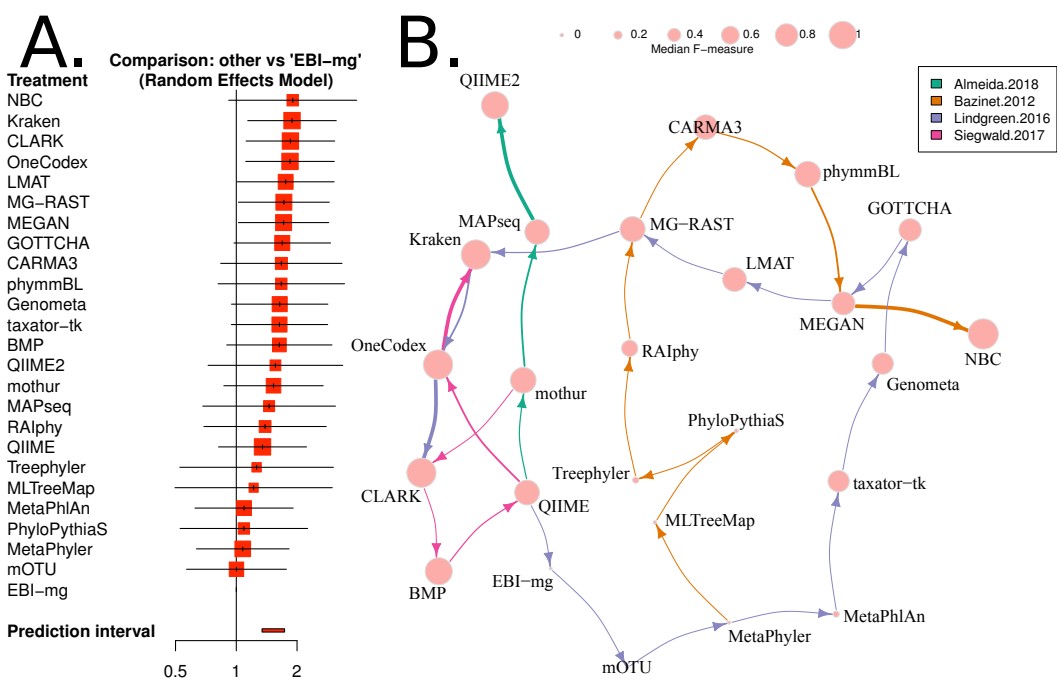

**Figure 4** **Network based comparison of metagenome analysis software tools.** (A) A forest plot of a network analysis, indicating the estimated accuracy range for each tool. The plot shows the relative *F*-measure with a 95% confidence interval for each software tool. The tools are sorted based upon relative performance, from high to low. Tools with significantly higher F-statistics have a 95% confidence interval that does not cover the null odds-ratio of 1. (B) A network representation of the software tools, published evaluations, ranks for each tool and the median *F*-measure. The edge-widths indicate the rank of a tool within a publication (based upon median, within-publication, rank). The edge-colours indicate the different publications, and node-sizes indicate median *F*-measure (based upon all publications). An edge is drawn between tools that are ranked consecutively within a publication.

the lowest-common-ancestor (LCA) algorithm for assigning read-similarities to taxonomy (*Huson et al., 2007*)

After manually extracting sensitivity, PPV and *F*-measures (or computing these) from the tables and/or Supplementary Materials for each publication (*Bazinet & Cummings, 2012*; *Lindgreen, Adair & Gardner, 2016*; *Siegwald et al., 2017*; *Almeida et al., 2018*), we have considered the within-publication distribution of accuracy measures (see Fig. 2C, Figs. S2 & S3). These figures indicate that each publication has differences in *F*-measure distributions. These can be skewed and multimodal, and different measures of centrality and variance. Therefore, a correction needs to be used to account for between-benchmark variation.

Firstly, we use a non-parametric approach for comparing corrected accuracy measures. We converted each *F*-measure to a "robust *Z*-score" (see 'Methods'). A median *Z*-score was computed for each software tool, and used to rank tools. A 95% confidence interval was also computed for each median *Z*-score using a bootstrapping procedure. The results are presented in Fig. 3B (within-benchmark distributions are shown in Fig. S3).

The second approach we have used is a network meta-analysis to compare the different results. This approach is becoming widely used in the medical literature, predominantly as a means to compare estimates of drug efficacy from multiple studies that include different cohorts, sample sizes and experimental designs (*Lumley, 2002*; *Lu & Ades, 2004*; *Salanti et al., 2008*; *Higgins et al., 2012*; *Greco et al., 2015*). This approach can incorporate both direct and indirect effects, and incorporates diverse intersecting sets of evidence. This means that indirect comparisons can be used to rank treatments (or software tool accuracy) even when a direct comparison has not been made.

We have used the "netmeta" software utility (implemented in R) (*Rücker et al., 2015*) to investigate the relative performance of each of the 25 software tools for which we have data, using the *F*-measure as a proxy for accuracy. A random-effects model and a rank-based approach were used for assessing the relative accuracy of different software tools. The resulting forest plot is shown in Fig. 4A.

The two distinct approaches for comparing the accuracies from diverse software evaluation datasets resulted in remarkably consistent software rankings in this set of results. The Pearson's correlation coefficient between robust $Z$-scores and network meta-analysis odds-ratios is 0.91 ($P$-value $= 4.9 \times 10^{-10}$), see Fig. S6.

## CONCLUSIONS

The analysis of environmental sequencing data remains a challenging task despite many years of research and many software tools for assisting with this task. In order to identify accurate tools for addressing this problem a number of benchmarking studies have been published (*Bazinet & Cummings, 2012*; *Peabody et al., 2015*; *Lindgreen, Adair & Gardner, 2016*; *Siegwald et al., 2017*; *McIntyre et al., 2017*; *Sczyrba et al., 2017*; *Almeida et al., 2018*). However, these studies have not shown a consistent or clearly optimal approach.

We have reviewed and evaluated the existing published benchmarks using a network meta-analysis and a non-parametric approach. These methods have identified a small number of tools that are consistently predicted to perform well. Our aim here is to make non-arbitrary software recommendations that are based upon robust criteria rather than how widely-adopted a tool is or the reputation of software developers, which are common proxies for how accurate a software tool is for eDNA analyses (*Gardner et al., 2017*).

Based upon this meta-analysis, the k-mer based approaches, CLARK (*Ounit et al., 2015*), Kraken (*Wood & Salzberg, 2014*) and One Codex (*Minot, Krumm & Greenfield, 2015*) consistently rank well in both the non-parametric, robust $Z$-score evaluation and the network meta-analysis. The confidence intervals for both evaluations were comparatively small, so these estimates are likely to be reliable. In particular, the network meta-analysis analysis showed that these tools are significantly more accurate than the alternatives (i.e., the 95% confidence intervals exclude the the odds-ratio of 1). Furthermore, these results are largely consistent with a recently published additional benchmark of eDNA analysis tools (*Escobar-Zepeda et al., 2018*).

There were also a number of widely-used tools, MG-RAST (*Wilke et al., 2016*), MEGAN (*Huson et al., 2016*) and QIIME 2 (*Bokulich et al., 2018*) that are both comparatively user-friendly and have respectable accuracy ($Z > 0$ and narrow confidence intervals, see Fig. 3B

and Fig. S5). However, the new QIIME 2 tool has only been evaluated in one benchmark (*Almeida et al., 2018*), and so this result should be viewed with caution until further independent evaluations are undertaken. Therefore QIIME2 has a large confidence interval on the accuracy estimate based upon robust $Z$-scores (Fig. 3) and ranked below high-performing tools with the network meta-analysis (Fig. 4). The tools Genometa (*Davenport et al., 2012*), GOTTCHA (*Freitas et al., 2015*), LMAT (*Ames et al., 2013*), mothur (*Schloss et al., 2009*) and taxator—tk (*Dröge, Gregor & McHardy, 2015*), while not meeting the stringent accuracy thresholds we have used above were also consistently ranked well by both approaches.

The NBC tool (*Rosen, Reichenberger & Rosenfeld, 2011*) ranked highly in both the robust $Z$-score and network analysis, however the confidence intervals on both accuracy estimates were comparably large. Presumably, this was due to its inclusion in a single, early benchmark study (*Bazinet & Cummings, 2012*) and exclusion from all subsequent benchmarks. To investigate this further, the authors of this study attempted to run NBC themselves, but found that it failed to run (core dump) on test input data. It is possible that with some debugging, this tool could compare favourably with modern approaches.

These results can by no means be considered the definitive answer to how to analyse eDNA datasets since tools will continue to be refined and results are based on broad averages over multiple conditions. Therefore, some tools may be more suited for more specific problems than those assessed in these results (e.g., human gut microbiome). Furthermore, we have not addressed the issue of scale—i.e., do these tools have sufficient speed to operate on the increasingly large-scale datasets that new sequencing methods are capable of producing?

Our analysis has not identified an underlying cause for inconsistencies between benchmarks. We found a core set of software tools that have been evaluated in most benchmarks. These are CLARK, Kraken, MEGAN, and MetaPhyler, but the relative ranking of these tools differed greatly between some benchmarks. We did find that restricting the included benchmarks to those that satisfy the criteria for a "neutral comparison study" (*Boulesteix, Lauer & Eugster, 2013*), improved the consistency of evaluations considerably. This may point to differences in the results obtained by "expert users" (e.g., tool developers) compared and those of "amateur users" (e.g., bioinformaticians or microbiologists).

Finally, the results presented in Fig. S4 indicate that most eDNA analysis tools have a high positive-predictive value (PPV). This implies that false-positive matches between eDNA sequences and reference databases are not the main source of error for these analyses. However, sensitivity estimates can be low and generally cover a broad range of values. This implies that false-negatives are the main source of error for environmental analysis. This shows that matching divergent eDNA and reference database nucleotide sequences remains a significant research challenge in need of further development.

## ACKNOWLEDGEMENTS

We are grateful to Jonathan Jacobs (@bioinformer) for maintaining a freely accessible list of metagenome analysis methods. We thank the authors of the benchmark studies

used for this work, all of whom were offered an opportunity to comment on this work before it was released. In particular: Adam Bazinet, Christopher E. Mason, Alexa McIntyre, Fiona Brinkman, Alice McHardy, Alexander Sczyrba, David Koslicki and Léa Siegwald for discussions and feedback on our early results. We are grateful to Anne-Laure Boulesteix for valuable suggestions on how to conduct the analyses.

### Funding
This work was the outcome of a workshop funded as part of the Biological Heritage National Science Challenge, New Zealand. Paul P. Gardner is funded by the Bioprotection Research Centre, the National Science Challenge ''New Zealand's Biological Heritage,'' and a Rutherford Discovery Fellowship. The funders had no role in study design, data collection and analysis, decision to publish, or preparation of the manuscript.

### Grant Disclosures
The following grant information was disclosed by the authors:
Biological Heritage National Science Challenge, New Zealand.
Bioprotection Research Centre, the National Science Challenge.
Rutherford Discovery Fellowship.

### Competing Interests
The authors declare there are no competing interests.

### Author Contributions
- Paul P. Gardner conceived and designed the experiments, performed the experiments, analyzed the data, contributed reagents/materials/analysis tools, prepared figures and/or tables, authored or reviewed drafts of the paper, approved the final draft.
- Renee J. Watson conceived and designed the experiments, performed the experiments, contributed reagents/materials/analysis tools, prepared figures and/or tables, authored or reviewed drafts of the paper.
- Xochitl C. Morgan conceived and designed the experiments, analyzed the data, prepared figures and/or tables, authored or reviewed drafts of the paper, approved the final draft.
- Jenny L. Draper, Robert D. Finn, Sergio E. Morales and Matthew B. Stott conceived and designed the experiments, analyzed the data, authored or reviewed drafts of the paper, approved the final draft.

### Data Availability
 GitHub: https://github.com/Gardner-BinfLab/meta-analysis-eDNA-software.

### Supplemental Information
Supplemental information for this article can be found online at http://dx.doi.org/10.7717/peerj.6160#supplemental-information.

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
