# Peer review of "Identifying accurate metagenome and amplicon software via a meta-analysis of sequence to taxonomy benchmarking studies"

_PeerJ, doi:10.7717/peerj.6160_

## Round 0.1 · original submission · Major Revisions

Dear authors,

Thank you for submitting this interesting manuscript to PeerJ.

Based on the reviews and my own opinions I have concluded that a major revision is required. Please take all the reviewers comments into account when revising the manuscript. The following two important issues are needs to be addressed in particular:

- In the conclusion, please make clearer recommendations on which tools to use, and explain in more detail how these stand apart from the rest. If such recommendations are difficult to make, please explain why and change the rest of the manuscript to reflect that.

- The introduction should concentrate on why and how tools should be evaluated and compared, and why a meta-analysis is necessary. The actual challenges met by the tools that have been benchmarked should be described in detail, while the general overview of microbiome sequencing and analysis in the introduction should be more succinct.

I look forward to see a revised version of the manuscript.

Sincerely,
Torbjørn Rognes

Reviewer 1 ·

Basic reporting

English used throughout is mostly fine. Would have the following comments:

1. Often the word “method” is thrown around too loosely to refer to the bioinformatics software being evaluated, where a better word would be “tool”, “software”, or “program”, to differentiate from the methods that these tools might actually implement, and which might be shared by more than one tool. Examples: twice in the abstract, last column header in Table 1, lines 100, 372, 404, and likely many other places.
2. Line 50, other places: “reads” are mentioned, though contigs and other types of sequences are possible. Consider replacing “reads” with “sequences”.
3. Line 74: “k-mers frequencies”: should be “k-mer frequencies”. Also check consistency of spelling and capitalization of “k-mer” throughout document, it is currently inconsistent.
4. Figure 1: “Metagenomic sequencing” is referenced in the second box from the top, though this is not a good descriptor (everything under discussion is metagenomic sequencing). Better would be to use the term used in the figure legend, “Shotgun metagenomics”.
5. Lines 111-112: sentence is not grammatically correct, please revise.
6. Lines 201-202: make “Pearson’s correlation coefficients” singular to agree with the other items in the series.
7. Line 208: not sure what a “taxonomic threshold” is in this context or how it can vary. Perhaps this should be described in more detail.
8. Line 218: “dependant” should be “dependent”.
9. Line 232: “GoogleScholar” should be two words.
10. Line 232: “7th July 2017” should be either “July 7th 2017” or perhaps better, “7 July 2017”. In any case, all dates referenced in the manuscript should be consistently formatted (according to journal standards, if there are any).
11. Line 249: the word “illustrated” would be better replaced with “shown”.
12. Line 250: “re-samplings” doesn’t need to be hyphenated.
13. Line 252: double space where there shouldn’t be. Other instances of this throughout manuscript as well, should check the whole thing.
14. Line 255: ‘metacont’ is in single quotes. I don’t see a need to sometimes use double quotes and sometimes use single quotes throughout the manuscript. I would pick one or the other and use it consistently. If you want to make function calls stand out as different from regular text, perhaps a different font can be used (e.g., Courier).
15. Lines 267-268: “we found that 45% … of all published tools have been evaluated”. Though the authors go on to say it, “evaluated” in this context really means evaluated to their standards. There are other instances in the manuscript where “evaluated” is used in this strict sense. I would make this very clear the first time “evaluated” is used this way, and include the additional explanation given in lines 269-277 as well. Then subsequently the reader will know what “evaluated” means without further explanation necessary.
16. Line 301: “the Sczyrba study” vs e.g., “Sczyrba et al.” elsewhere – studies should be referenced in a consistent manner throughout the manuscript. Also, format of “et al.” should be checked throughout the manuscript as sometimes it is italicized and sometimes it isn’t (whatever the journal standard is should be followed, I think not italicized is better English).
17. Line 302: extra space between the words “a much”
18. Line 338: “, stemming from a diverse intersecting sets of evidence”. This sentence needs to be revised to make sense.
19. Line 342: remove comma after “performance”
20. Line 343: unnecessary space in “random- effects”
21. Line 382: “sequence“K-mers””: needs a space
22. Line 413: comma needed after “i.e.”
23. Lines 413-415: sentence phrased as a question should end in a question mark.
24. Line 420: ampersand looks odd but maybe this is the journal standard.
25. Line 421: extra space after comma
26. Line 428: “higher taxonomic levels”: higher than what? Could be phrased more specifically.
27. Lines 433-435: “Spike-ins … could be very useful for this.” This is not a complete sentence, please revise.
28. I noticed Fig./Figure are sometimes used inconsistently.


Literature references are mostly adequate. However, I would recommend the following changes:

1. The metagenomic assembly references on lines 57-58 are not great. The 2008 review is before any sort of true metagenomic assemblers were available. The Nagaranjan and Pop reference is mostly about generic genome assembly, not necessarily metagenome assembly. And the Magoc et al. study is not really about metagenomics either. Better references would include the following, for example (and there may be others): (1) Breitwieser, Lu, and Salzberg (2017): A review of methods and databases for metagenomic classification and assembly. (2) Olson et al. (2017): Metagenomic assembly through the lens of validation: recent advances in assessing and improving the quality of genomes assembled from metagenomes.
2. Please add the MEGAN assembler (Huson et al. 2017) to the list of targeted gene assembly references, which currently includes SAT-assembler and Xander. MEGAN assembler was shown to have some performance advantages over Xander.
3. Very few specific references to metagenomic assembly tools are given. Currently, only reference-guided “gene-centric” or “targeted gene assembly” tools and references are given. No specific references to whole-genome metagenomic assemblers are given. Two of the best current de novo whole-genome metagenomic assemblers are MEGAHIT and metaSPAdes; these should be mentioned. There are also reference-guided whole-genome metagenomic assembly tools, such as MetaCompass, that could be mentioned. This current manuscript is not targeted at metagenomic assembly and currently does a poor job of characterizing this body of work. While I understand the authors are trying to describe the whole metagenomic workflow, either it needs to be done more thoroughly, or it should be streamlined so that steps are described more succinctly and generically.
4. I don’t think all of the tools listed in Figure 1 are cited in the manuscript; they should be (somehow/somewhere). For boxes that are given multiple times (e.g., taxonomic assignment and functional assignment), are the tools listed always specific to that “path” through the workflow, or are sometimes different tools listed arbitrarily? (I suspect the latter, which in my mind is somewhat misleading.)
5. Line 268: “Metagenomics – Tools, Methods and Madness”: I don’t really know what this refers to and does not look like a proper citation. Perhaps more info should be included in the text (author, URL, etc.).
6. Lines 348-349: citations for these software tools are missing here.


Field background/context provided:
In general, I think a decent effort is made to give background. However, the meta-evaluation in the paper concerns only a very specific part of a metagenomic workflow (taxonomic assignment), so much of the background (sketched out in Fig. 1) is not really necessary and better left to a review paper. Partly because of this, sections such as metagenomic assembly are not described very accurately or thoroughly. Because the core problem this paper is concerned with, taxonomic assignment, is in some ways independent of the workflow it’s embedded in (from a collection of sequences, either return a sequence-by-sequence taxonomic assignment or some percentage-based profile), a comprehensive description of all possible metagenomic workflows in which taxonomic assignment is performed is not really necessary. On the plus side, background concerning taxonomic assignment specifically is good: Table 1 is mostly very good, the section on positive and negative control dataset selection is good, and issues regarding metrics used for benchmarking are treated well. Here are some specific recommendations:
1. I think the current description of the global metagenomic workflow (i.e., Figure 1 and associated text) should be streamlined and shortened, but alternatively it could be fleshed out (I gave ideas regarding how the metagenomic assembly section could be improved, for example).
2. Table 1: I think the “Positive Control” column descriptions could be described more consistently. A bolded number draws your attention (e.g., 742, 417, 846, 11, 689, 125) but these refer variously to taxa, genera, species, and other things. Perhaps more effort could be made to describe the data analyzed in these studies in such a way that would facilitate easy comparison. Also, capitalization on Table 1 column headers should be made consistent.
3. Figure 2: I only just noticed the subtle difference between the “Simulated evolution” diagram and the “New genome sequences” diagram (aside from the color of the dotted lines), namely that “Simulated evolution” could extend from a tip on phylogeny whereas a new genome sequence wouldn’t. I think this should be made more obvious.
4. Lines 182-197: I would lead with the two major analysis paradigms (distributional recapitulation and read-by-read), and then talk about implications for definitions of TP/FP/TN/FN in each of these contexts.
5. Line 190: “This approach”: starting a new paragraph this way, I’m not sure which approach is being referred to.

The article structure, figures, and tables are mostly fine. I would perhaps put less effort into trying to describe metagenomic workflows in all their possible permutations. Furthermore, there is some information in the Conclusions that is not really germane to the study at hand or supported by the results, namely the final paragraph on spike-ins. I did not see it mentioned anywhere in the results that comparative evaluation studies that used negative controls were more fair, produced better results, etc. Perhaps this paragraph should be incorporated into the introduction/background material instead.

Experimental design

This is original primary research within the scope and aims of the journal.

The research question is well defined and fills an obvious knowledge gap (conflicting reports between different comparative evaluation studies).

I believe the meta-comparison was done rigorously and ethically, though I have a few concerns:
1. The rankings in Figures 4 and 5 are currently by median F-measure. However, the confidence interval is not currently taken into account in the ranking. For example, in Fig. 4, TACOA has a huge confidence interval whereas the three tools below it (DiScRIBinATE, commonkmers, and CARMA3) have almost the same median F-measure but much tighter confidence intervals reflecting the fact that they perform more consistently and/or were evaluated more thoroughly. Based on this data I would be inclined to choose any of tools ranked #2, #3, or #4 over the tool ranked #1. Might it be possible to come up with some ranking scheme that is a composite of median F-measure and size of confidence interval? This is important for Figure 4, anyway; the confidence intervals in Fig. 5 are all pretty darn big and all about the same size, making me question how much real information is contained in that analysis.
2. A very big issue is how the five “consistently highly ranked” performers (CARMA3, DiScRIBinATE, MetaPhlan2.0, OneCodex, and TACOA) were chosen. These represent five of the top seven tools in Fig. 4, and five of the top nine tools in Fig 5. However, by this criterion (ranking within the top nine in both analyses, say), commonkmers, DUDes, and MetaBin should also be included. This is important because people are going to remember the tools that were stated by name to be top performers, and it would seem that these five were arbitrarily selected. Not only is the cutoff (top nine) arbitrarily selected, but three tools that make the cut in both analyses were omitted. I think a more principled way of selecting top performers is needed. For example, look for a break in each distribution or state explicitly where the cutoff is (top five, top 10, etc.), and then name all the tools that make the cut in both analyses, not just some of them. Moreover, see my previous point about how the rankings might change if confidence interval were somehow factored in. This has lots of implications because much of the ensuing discussion treats the five selected tools specifically.
3. There is obvious concordance between the lists resulting from the two ranking approaches (Figures 4 and 5) as they’re currently ordered; however, it might be nice to quantify this somehow (e.g., calculate “mean rank discrepancy”, i.e., what is the average difference in rank for a given tool between the two approaches?)

Methods are described in sufficient detail to replicate, especially since the scripts used to scrape data from the existing studies are provided via GitHub.

Validity of the findings

I find the methods applied to be appropriate and statistically sound. However, I take issue with some of the conclusions (specifically, the five tools selected as top performers), as noted in my previous comments.

Additional comments

Overall, I like this study --- it’s a good concept, the stats developed and methods of analysis used seem appropriate, there are some very nicely drawn figures, etc. I feel like the manuscript could be improved, however, by focusing more specifically on taxonomic assignment (much of the background or supporting information may be interesting, but isn’t necessarily relevant to the results or conclusions of the present study – e.g., information about metagenomic workflows generally). Also, the discussion of negative controls, types of metrics, potential usefulness of spike-ins, etc. is interesting and useful information, but not really necessary in order to understand the results of the study. The manuscript is a little weird in that respect --- it feels like a hybrid review paper/new results paper. Also, and most importantly, I think the rankings and subsequent selection of “top performers” needs to be reevaluated, as this takeaway is what most people will be interested in.
Here are some additional specific comments:
1. Abstract, line 21, and main text, line 98: should also talk about robust Z-scores in addition to network meta-analysis. I actually think the robust Z-scores approach contains more data and is likely to be more informative than the network meta-analysis, and is weighted just as heavily in the conclusions drawn.
2. Line 231: should there be a link to the project referenced?
3. Lines 310-311: “The different software versions do not explain this discrepancy.” How do you know this? What sorts of analyses were performed to determine this?
4. Figure 4A: in the legend, circle size is defined as “proportional to median F-measure value”; at the top of the figure, circle size is defined as “Number of F-measure estimates”. These sound like different things to me, I think this definition needs to be made clearer.
5. Lines 350-352: “small numbers of direct comparisons between these methods” --- does this mean direct comparisons between these methods and something else? If so, what is that “something else”? Or do you mean small numbers of direct comparisons among methods (either these five or all methods more generally)?
6. Supplementary figures 1 and 3 are never referenced in the main text.

In summary, I enjoyed reading this manuscript and I think the data analysis could eventually be useful to the community. I encourage the authors to spend time revising the manuscript, think about how the findings can be best presented in context (and providing only the most relevant context), and be very careful about how the findings are interpreted. In particular, I worry a little bit about the amount of information contained in the network analysis given the consistently large size of the confidence intervals in Fig. 5.

Reviewer 2 ·

Basic reporting

The paper is generally clear. There are some problems with the structure as I describe below.

Experimental design

The experimental design is generally sound. There are some inconsistencies as i describe in my comments below

Validity of the findings

The paper fails to achieve a real conclusion. The conclusion can probably be either the recommendation of a tool (or set of tools) to profile microbial communities, or the striking inconsistency of available evaluations. The main conclusion can only be one of the two, but at the moment it is unclear what is the conclusions of the paper.

Additional comments

- Given the title of the paper and several of its paragraphs (incl "Our aim here is to make non-arbitrary software recommendations"), the reader will expect to find in the paper a recommendation based on the evaluation meta-analysis about which computational tools for his/her research. However, the work does not provide any real advice for this but it rather push the message that the evaluations are inconsistent and that no "winner" can be established. I think the paper should be either restructured to provide clear recommendations on the methods to use, or it should focus (starting from the title) on the message that the evaluations are inconsistent.

- The statistics in Figure 3A and Figure 3B are interesting and provide a nice summary of the available methods. It would be even more interesting if the statistics (specifically citations) of Figure 3B are correlated with the actual performances in the evaluations.

- The paragraph between lines 370 and 390 is weird to me. Here you describe how some methods work, which is disconnected with the rest of the results. I think this part should be removed and replaced with a better attempt at interpreting the results of the meta-analysis.

- I think that also the intro is a bit weird because it reports a mini-review of what amplicon sequencing and metagenomics are, whereas the focus of the paper is on a meta-analysis of microbiome profiling tools. I think the authors should focus the intro on the need to comparatively evaluate computational profiling tools and what are the challenges in doing it. There are many reviews of the microbiome sequencing field you can just cite for the interested reader.

- I very much agree with the three listed in the section "Overview of eDNA analysis evaluations". However, principle number 2 is not satisfied by most of the papers used in the meta-analysis. I think this is a very crucial point. Actually, it would be very interesting to highlight the performance of a tool in the evaluation in which the developers of the tool are in the author list with respect to the evaluations in which the developers of the tool are not in the author list. In any case, the authors need to be more consistent: if they truly believe that principle number 2 is right (and I fully agree with that), then they have to explain why they still use evaluations breaking this principle.

- I could be wrong, but the evaluation by McIntyre et al (for which a journal publication is available and should replace the bioRxiv citation) includes datasets used by other studies, and maybe even other considered evaluations. This would not be ideal and should be discussed. Again, I could be wrong here, but please double check.

- The use of "eDNA" is not standard and I think should be avoided. If you want a term covering both shotgun sequencing and amplicon sequencing you can just use terms like "microbiome sequencing"

Figure 1. There are several inconsistencies in this figure:
- The "metagenomic reads" to "binning" link misses the metagenomic assembly part which is instead mentioned downstream of the binning. This is incorrect and misleading. CONCOCT and MetaBat requires contigs (and reads) as input, not binned reads only.
- not sure what "amplicon contigs" mean? If you refer to merging the paired end reads for the amplicon, it is not very correct to call it "assembly"
- closed-reference OTU assignment does not need OTU clustering as starting point
- I would not call PICRUST a functional assignment tool. Maybe "functional potential prediction tool"?

Introduction:
- when introducing metagenomic assembly, the authors do not cite the state-of-the-art tools (as reported by some of the benchmark papers they consider later on in the paper)

---

## Round 0.2 · Minor Revisions

Thank you for submitting a revised and clearly improved manuscript.

Please respond to the comments by the reviewer, including those in the attached annotated PDF.

Regarding figure 4: The reviewer noted that the colours used for the different studies in this figure differ from those used in the other figures. I noted also that the study by Almeida et al. (2018) is not included in this figure. It seems to me like an old version of this figure was used in the submission as you have included a newer version of the figure in one of the other files submitted. The new version uses other colours and also includes the study by Almeida et al. (2018). Please include the correct version of this figure with your resubmission.

Reviewer 1 ·

Basic reporting

The English used throughout the manuscript is fine, although there were many more grammatical mistakes, formatting issues, etc. than I would have liked to have dealt with as a reviewer.

Experimental design

No comment

Validity of the findings

No comment

Additional comments

All-in-all, the manuscript is much improved — in particular by removing much of the attempt at a comprehensive review of metagenomics workflows, and by improving the methods used to select well-performing tools (and subsequent discussion of those tools).

I’m attaching a combined annotated PDF of the main manuscript and the supplementary material that the authors should review carefully. I’ll reiterate a couple of the more important points here, though.

1) The F1 measure fundamentally weights sensitivity and PPV equally. Thus, tools that have been developed with high PPV in mind (not necessarily sensitivity), such as marker gene-based approaches, suffer using the current ranking methodology. Indeed, it’s no surprise that the three top-ranked tools by PPV in Fig S5 are all marker gene-based (GOTTCHA, MetaPhlAn, and mOTU). I think the authors need to make it more clear that tools that are not "balanced" with regard to the sensitivity/PPV tradeoff are not going to do well in their ranking, and they may wish to call out top-performing tools (either by sensitivity or PPV) in their conclusion. They touch on this to some extent in the final paragraph of the conclusion, but I don’t think it’s explicit enough with regard to the points that need making here.

2) The colors used to represent the various benchmark studies in Fig 4 are different from those in other figures — confusingly so. This should be corrected for consistency.

Annotated reviews are not available for download in order to protect the identity of reviewers who chose to remain anonymous.

---

## Round 0.3 · accepted · Accept

I think the authors have made the changes necessary to warrant publication.

#